# The Dependence of Running Speed and Muscle Strength on the Serum Concentration of Vitamin D in Young Male Professional Football Players Residing in the Russian Federation

**DOI:** 10.3390/nu11091960

**Published:** 2019-08-21

**Authors:** Eduard Bezuglov, Aleksandra Tikhonova, Anastasiya Zueva, Vladimir Khaitin, Anastasiya Lyubushkina, Evgeny Achkasov, Zbigniew Waśkiewicz, Dagmara Gerasimuk, Aleksandra Żebrowska, Pantelis Theodoros Nikolaidis, Thomas Rosemann, Beat Knechtle

**Affiliations:** 1Department of Sport Medicine and Medical Rehabilitation, Sechenov First Moscow State Medical University, 119435 Moscow, Russia; 2Department of Sport Medicine and Medical Rehabilitation, Faculty of Continuing Professional Education, Sechenov First Moscow State Medical University, 119435 Moscow, Russia; 3FC Zenit Saint-Petersburg, 197341 Saint Petersburg, Russia; 4«Smart Recovery» Sports Medicine Clinic, 121552 Moscow, Russia; 5Institute of Sport Science, Jerzy Kukuczka Academy of Physical Education, 40-065 Katowice, Poland; 6Department of Sports Training, Jerzy Kukuczka Academy of Physical Education, 40-065 Katowice, Poland; 7Department of Physiological and Medical Sciences, Jerzy Kukuczka Academy of Physical Education, 40-065 Katowice, Poland; 8Exercise Physiology Laboratory, 18450 Nikaia, Greece; 9School of Health and Caring Sciences, University of West Attica, 11244 Athens, Greece; 10Institute of Primary Care, University of Zurich, 8091 Zurich, Switzerland; 11Medbase St. Gallen Am Vadianplatz, 9001 St. Gallen, Switzerland

**Keywords:** vitamin D_3_, cholecalciferol, muscle power and speed, vitamin D deficiency, treatment for vitamin D deficiency, young football players

## Abstract

*Background*: Vitamin D insufficiency is prevalent among athletes, and it can negatively affect physical performance. At the same time, most of the available data were obtained from untrained individuals of various ages, and published studies performed in athletes led to contradictory conclusions. *Methods*: This cohort prospective study examined the serum concentration of 25-hydroxycalciferol (25(OH)D) and its association with running speed and muscle power in 131 young football players (mean age 15.6 ± 2.4 years). *Results*: 25(OH)D levels were below reference in 42.8% (serum 25(OH)D <30 ng/mL) and above reference in 30.5% of the participants (serum 25(OH)D 61–130 ng/mL). A comparison of the results of 5, 15, and 30 m sprint tests and the standing long jump test found no statistically significant differences between the two groups. Athletes from the 25(OH)D-insufficient group were treated with 5000 IU cholecalciferol supplement daily for 60 days. After the treatment, the 25(OH)D concentration increased by 79.2% and was within reference in 84% of the treated athletes (serum 25(OH)D 30–60 ng/mL). Testing was repeated after the end of treatment, and a statistically significant increase in the results of the 5, 15, and 30 m sprint tests was observed (Cohen’s *d* was 0.46, 0.33, and 0.34, respectively), while the results of the standing long jump test remained unchanged. Body height, body weight, and lean body mass of the football players also increased. *Conclusions*: These findings indicate that there is likely no correlation between serum levels of 25(OH)D, muscle power, and running speed in young professional football players, and the changes observed post-treatment might have been caused by changes in the anthropometric parameters. During the study, all the anthropometric parameters changed, but the amount of lean body mass only correlated with the results of the 5 m sprint.

## 1. Introduction

Vitamin D plays a crucial role in phosphorus and calcium metabolism and thus affects the state of bone tissue. Its active form, 1,25-dihydroxyvitamin D (1,25(OH)2D), increases the efficiency of intestinal calcium absorption from 10%–15% to 30%–40% by interacting with the vitamin D receptor and retinoid X receptor (VDR-RXR), thereby promoting the expression of an epithelial calcium channel and a calcium-binding protein [1,2,3]. It has been estimated that 1,25(OH)2D also increases intestinal phosphorus absorption from 50%–60% to approximately 80% [4,5]. It promotes resistance to certain diseases and affects the immune system as well as maintains muscle tone and the structure of connective tissue. It also regulates lipid and carbohydrate metabolism and the level of blood glucose [6,7,8]. Vitamin D receptors can be found in a multitude of tissues, which explains its numerous effects outside the skeletal system [9,10].

Vitamin D insufficiency is most prevalent in regions located to the north of the 35th parallel north because sun rays enter the atmosphere at a shallower angle and disperse [11,12]. Most of the vitamin D found in human body is synthesized when ultraviolet (UV) rays penetrate the open skin at a specific angle. Two distinct forms of vitamin D exist: ergocalciferol (vitamin D_2_), which is primarily obtained from plant-based food, and cholecalciferol (vitamin D_3_), which is synthesized when the body is exposed to UV rays. The vitamin D_3_ metabolite 25-hydroxycalciferol (25(OH)D) is an important biomarker used in clinical settings to prevent and treat vitamin D deficiencies [13].

Vitamin D insufficiency is also very common in professional athletes, where it can reach 60%–90%, according to a number of different authors [14,15,16,17]. Vitamin D deficiency presents an important challenge in football, where it was observed in 64%–83% of football players from England, Spain, and Poland [18,19]. However, as shown by Hamilton et al., it is most frequent in the Middle East, where it was diagnosed in 84% of the 342 examined Qatari football players [20].

Vitamin D has been shown to affect muscle tissue, which serves as an important target site. However, most studies confirming the link between vitamin D deficiency and muscle weakness were performed on people of varying ages with no adequate training [17,18,19,20,21,22,23,24,25,26]. This link may be facilitated through multiple pathways, which either directly affect muscle tissue or, possibly, alter endogenous testosterone synthesis. Pilz et al. showed that vitamin D exhibits ergogenic potential and indirectly enhances testosterone production, which can also affect the muscular system. It is possible that this is achieved through inhibition of testosterone aromatization and enhanced binding of androgens, which in turn leads to muscle hypertrophy and increase in strength [27]. Animal studies have shown that vitamin D influences myostatin inhibition, regeneration, and muscle cell proliferation processes [28,29].

Notably, most of the existing studies exploring the association between serum vitamin D level and muscle performance in athletes were performed on adults and yielded contradictory results. A systematic review by Chiang et al. showed that the administration of vitamin D_3_ supplements achieved a statistically significant improvement in muscle performance, which was not obtained when vitamin D_2_ supplements were used [30]. van Hurst et al. noted that muscle strength and stamina associated with vitamin D administration only occurred in athletes whose vitamin D levels were initially low [31]. Farrokhyar et al. found no association between vitamin D supplementation, vitamin D concentration, and various indicators of physical performance, including muscle strength [32].

Football players have also been the subjects of such studies. Koundourakis et al. observed a positive correlation between vitamin D level and muscle performance in a cohort of Greek football players [33]. A randomized study performed by Close et al. also showed the beneficial effects of vitamin D on muscle strength and power, with athletes who had received 5000 IU vitamin D for eight weeks significantly improving their results in sprint and vertical jump tests [14]. Hamilton et al., on the other hand, found no significant association between the level of 25(OH)D and muscle function [20]. In a study by Jastrzebska et al., where 5000 IU vitamin D were administered to football players, most of the changes in the indicators of physical performance were insignificant [34].

So far, there have been a few studies regarding the prevalence of vitamin D deficiency in young athletes and its effect on their muscle performance. Brannstrom et al. observed 19 young female football players and found no statistically significant correlations between vitamin D levels and most of the indicators of muscle tissue performance [35]. A study by Fitzgerald et al. found that vitamin D insufficiency was highly prevalent in a cohort of 53 Canadian junior hockey players. However, no correlation between vitamin D levels and muscle strength was observed [36].

Thus, there is no consensus regarding the effect of serum concentration of vitamin D on running speed and strength in athletes. At the same time, most of the existing studies were performed in adult populations, with only occasional studies performed on young athletes. To the best of our knowledge, there is no published research on this topic that has been performed in a cohort of young football players, which increases the importance of studying the effects exerted by vitamin D insufficiency on muscle tissue. Therefore, the present study examined the serum concentration of vitamin D (25(OH)D) and its association with running effect and muscle power in young male football players.

## 2. Materials and Methods

The study was conducted from December 2018 to February 2019 at the Lokomed Medical Center of Lokomotiv FC Moscow with the participation of the staff of the Department of Sports Medicine and Medical Rehabilitation of the Sechenov First Moscow State Medical University. The protocol of this study was approved by the official Local Ethics Committee of the Sechenov First Moscow State University under the statement number 11–19 of 07/25/2019. All stages of the study complied with the legislation of the Russian Federation. All participants in the study provided their informed consent. Consent from the parents of all study participants under 18 years of age was obtained. Athletes who were 18 years or older provided the consent form directly.

This study summarizes the data obtained from a cohort of 131 white male football players from Football School Lokomotiv and FC Lokomotiv Moscow Youth team aged 12 to 23 years (mean age 15.6 ± 2.4 years, mean height 172.2 ± 9.9, mean weight 62.1 ± 10.9, mean body fat % 15.6 ± 3.7) who did not have any contraindications for sports. The study included young football players who trained at Football School Lokomotiv or FC Lokomotiv Moscow Youth team and permanently resided in Moscow, a city located at latitude 55° north.

### 2.1. Criteria for Exclusion from the Study

The criteria for exclusion from the study were as follows:-The athlete received vitamin D supplements 30 days or less prior to first blood sampling.-The athlete suffered from acute respiratory viral infections or any other condition that resulted in absence from three or more training sessions 30 days or less prior to the examination.-The athlete could not maintain daily contact with the medical personnel distributing vitamin D_3_ supplements.-The athlete spent more than three days outside Moscow during the last three months.-The athlete was expelled from the academy during the study.-The athlete refused to take part in speed and power testing.

### 2.2. Laboratory Testing

Two blood samples were obtained: the first in December 2018 and the second 70 days later. Overall, treatment was administered over 60 days with two five-day breaks after the end of the first and the second months.

Fasting blood samples were collected from the cubital vein in the morning. Two immunoassay blood tests for vitamin D_3_ (25(OH)D) were conducted using an in vitro reagent set for 25(OH)D produced by Euroimmun AG (Germany) and a Mindray MR-96A microplate reader (China). In accordance with the modern guidelines, vitamin D plasma concentrations below 30 ng/mL were considered insufficient (with 21–29 ng/mL and <21 ng/mL used as diagnostic criteria for vitamin D insufficiency and deficiency, respectively). Values within the range of 30–60 ng/mL were considered normal and values >60 ng/mL were considered as simply higher than normal indicators. Based on the results of the 25(OH)D blood test, groups of athletes with insufficient (group 1) and higher than normal (group 2) vitamin D were formed.

Body height and body weight measurements were obtained from all participants. In the groups with insufficient or excessive vitamin D, muscle and body fat mass measurements were obtained using bioimpedance analysis on the day following the first and the second blood sampling procedures. The ABC-02 “MEDASS” (Russia) analyzer was used for bioimpedance analysis. The procedure was performed in the morning before and after the treatment, with the patient in the fasting state. Running speed and power tests, i.e., 5, 15, and 30 m sprint tests and a standing long jump test, were performed in both groups. All the football players had previously repeatedly performed these tests at least 3–4 times within 2–3 years and were well acquainted with the rules for their conduct. After the warm-up, the long jump test was performed, followed by the sprint tests with 5 min breaks between each sprint. For the sprints, a timing system produced by Brower Timing Systems (USA) was used. Standing long jump distance was measured using a PLR 15 Digital Laser Measure produced by Bosch (Malaysia).

### 2.3. Description of the Sprint Tests

Each athlete started the sprint from a stationary standing position, with the leading (the nearest leg) foot located 20 cm before the starting line. Two pairs of timing gates had been set up: the first at the starting line and the second on the finish line. The athlete began the sprint at will. The results were immediately transferred from the timing gates to the chronometer and saved.

### 2.4. Description of the Standing Long Jump Test

Each athlete started the test with the legs shoulder-width apart, feet parallel, and arms by their side. Mid-flight, the athlete pulled their legs close to their body and extended them forward heels first, landing on both feet simultaneously. The length of the jump was measured along the perpendicular line from the point of push-off to the athlete’s heel upon landing. None of the athletes or personnel performing the tests knew which group the athlete was in.

### 2.5. Supplementation with Vitamin D

After the initial testing, athletes belonging to group 1 began receiving vitamin D correction therapy. Vitamin D deficiency and insufficiency were treated with 5000 IU oral cholecalciferol (SiS vitamin D_3_ 5000 IU, United Kingdom) daily after breakfast. Treatment lasted for 60 days, with a 5-day break after the 30th day of treatment, and was supervised daily by the medical staff of the Football School. After the end of treatment, a second series of tests (i.e., 5, 15, and 30 m sprint tests and the standing long jump test) was carried out in group 1, and a second set of bioimpedance measurements was obtained.

### 2.6. Statistical Analysis

IBM SPSS Statistics software v.23.0 (IBM, New York, NY, USA) was used for statistical analysis. Descriptive statistics and Kolmogorov–Smirnov test was used to determine the normality of the distribution. Student’s *t*-test for independent samples was used to compare weight, height, body mass index (BMI), body fat mass percentage, and 15 m sprint in groups 1 and 2. Mann–Whitney test for independent samples was used to compare lean body mass, 5 and 30 m sprints, and standing long jump in groups 1 and 2. Student’s *t*-test for dependent samples was used to compare weight, height, BMI, body fat mass percentage, and 30 m sprint before and after treatment. Wilcoxon signed-rank test was used to compare 5 and 15 m sprints, standing long jump, and lean body mass before and after treatment.

## 3. Results

Vitamin 25(OH)D levels were below reference values in 42.8% (56) of the examined football players. Vitamin 25(OH)D deficiency and insufficiency was observed in 19.9% (26) and 22.9% (30) of the players, respectively. In 57.2% (75) of the examined young football players, vitamin D levels were normal (in the range of 30–60 ng/mL in 26.7% (35) of players) or high (in the range of 61–130 ng/mL in 30.5% (40) of players) (Figure 1).

Based on the results of the vitamin D blood test, groups with insufficient (serum vitamin D < 30 ng/mL, group 1) and higher than normal (serum vitamin D > 60 ng/mL, group 2) vitamin D were formed. During the study, the array of tests (i.e., sprint tests and standing long jump test) was completed in full by 25 football players from each of the groups. Group 1 was composed of 25 individuals (mean age 13.96 ± 1.4 years) with the mean serum vitamin 25(OH)D level of 20.7 ng/mL. Group 2 was composed of 25 individuals (mean age 14.8 ± 1.6 years) with the mean serum vitamin 25(OH)D level of 84.5 ng/mL. Age, body mass, height, BMI, body fat mass, and body muscle mass were comparable in both groups (Table 1).

Speed and power tests were conducted in both groups. No statistically significant difference was found in the results of any of the tests (Table 2). After the 60 day vitamin D supplementation therapy course finished, mean vitamin 25(OH)D concentration in members of group 1 increased by 79.2% (from 20.7 to 31.7 ng/mL, *p* < 0.001), and reference values were achieved in 84% (21) of them. A repeated measurement of height, weight, body composition, and performance in speed and strength tests was carried out in group 1.

A statistically significant improvement in sprint results was observed (Table 3), and there were statistically significant increases in height, weight, and BMI (Table 4).

Thus, increasing blood level of 25(OH)D in the group of players with an initially insufficient level was associated with a statistically significant increase in performance in 5, 15, and 30 m sprint tests. These changes might be associated with not only an increased level of serum concentration (OH) D but also with a change in anthropometric indicators. However, we did not reveal a correlation between changes in height and weight and sprint rates. We studied the correlation between changes in lean body mass and changes in sprints. The correlation of lean body mass and the result of the 5 m sprint was the only result that was significant, i.e., the greater the change in lean body mass, the longer the time taken for running 5 m. Furthermore, in the 5 m run, there was a tendency for a correlation between changes in weight and height. Other indicators did not significantly correlate with sprint results (Table 5).

## 4. Discussion

The aim of the present study was to examine the serum concentration of vitamin D and its effect on running speed and muscle power in young male football players. The most important finding was that vitamin D concentration was below normal in a substantial share (42.8%) of youth professional football players residing in Moscow (latitude 55.9° north). However, previously published studies on vitamin D in football players of various ages and sex have reported a higher prevalence of vitamin D insufficiency and deficiency, even in regions with sufficient insolation [17,33,34,35]. The relatively low prevalence observed in Russian youth football players during the winter season may be explained by the lower workload compared to the workloads endured in summer and autumn and by the constant supervision by both coaches and doctors. Excessive exercise has been reported as a possible reason for a decrease in serum vitamin D level [37].

In the study, not all athletes with a low vitamin D content had the same response to vitamin D supplementation, which could be due to several factors. One of them is probably the uneven sensitivity of individuals to vitamin D, possibly due to molecular factors [38]. However, even a sufficient concentration of vitamin D may not indicate the expected effect on various body functions as these effects may depend on the content of other biologically active substances in the body, such as magnesium [39].

So far, the results of research examining the association between serum vitamin D concentration and muscle strength and running speed in athletes have been contradictory. A number of authors have found no evidence of any significant impact on various indicators of physical performance. Brännström et al. measured parameters of both speed and power in a cohort of 19 young female football players from Sweden, a region where insolation is limited. They found no significant correlation between these parameters, including jump and sprint performance, and vitamin D levels [35]. Similar results were obtained by Jastrzębska et al., who studied the effect exerted on various performance indicators by increasing serum vitamin D in a cohort of trained football players. The athletes assigned to the treatment group were administered 5000 IU vitamin D daily. No significant difference in performance improvement was observed between the control group and treatment group, even though the serum vitamin D level increased by 119.2% in the treatment group [34]. Fitzgerald et al. analyzed the blood test results and performance in the vertical jump test (power) and Wingate test (anaerobic power and capacity) in 53 young hockey players residing and training at latitude 44.9° north. They observed a statistically significant positive correlation between serum vitamin D concentration and the performance of these athletes [40]. Koundourakis et al. worked with a cohort of 67 professional Greek football players and reported that serum levels of vitamin D were directly associated with the performance in jump tests and in both the 10 and 20 m sprint tests. Their results showed that vitamin D levels were significantly associated with muscle power and speed as well as sprint performance and VO_2_ max in professional football players regardless of their performance level [33].

In our study, no statistically significant difference was observed between muscle power and running speed in groups with insufficient and excessive vitamin D despite the large gap in vitamin D concentrations between the two groups (20.7 and 84.5 ng/mL, respectively). After 60 days of treatment, a significant increase in both serum vitamin D levels and 5, 15, and 30 m sprint performance was achieved in the initially insufficient group. The improvement in performance might be associated with the increase in vitamin D levels. It should be noted, however, that the participants became taller and heavier during the study, and this anthropometric change may have also been the cause of the observed improvement in performance. Previous research on the association between increased vitamin D levels and both speed and strength performance did not consider a potential change in anthropometric parameters of the participants.

There is no “gold standard” that can exactly estimate the concentration of vitamin D biochemical markers in the human body. Most often, a serum concentration of 25(OH)D, which is the sum of 25(OH)D_3_ and 25(OH)D_2_, is used for this purpose, although a number of other biochemical agents have been proposed in recent years [38]. At the same time, isotope dilution liquid chromatography mass spectrometry can be considered the most accurate method for determining the biochemical markers of vitamin D; however, current immunoassays have demonstrated acceptable performance [41]. This hypothesis is supported by the fact that the tests performed initially did not find any significant difference between the performance of athletes from the insufficient and excessive vitamin D groups.

## 5. Conclusions

These findings indicate that there is likely no correlation between serum levels of 25(OH)D, muscle power, and running speed in young professional football players, and the changes observed post-treatment may have been caused by the changes in anthropometric parameters. During the study, all the anthropometric parameters changed, but the amount of lean body mass only correlated with the results of 5 m sprint.

## Figures and Tables

**Figure 1 nutrients-11-01960-f001:**
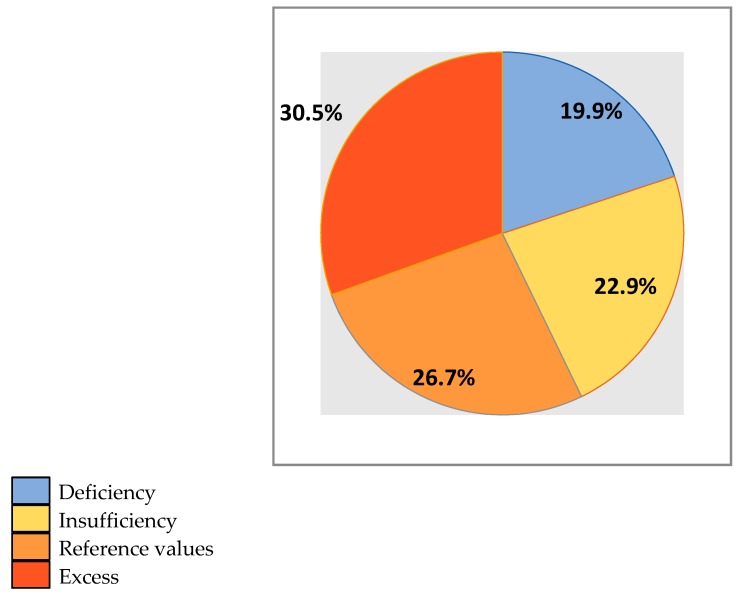
Serum level of vitamin 25-hydroxycalciferol (25(OH)D) in young professional football players permanently residing in Moscow (percentage).

**Table 1 nutrients-11-01960-t001:** Comparison of age, body weight (kg), body height (cm), body mass index (BMI), body fat mass percentage, and lean body mass percentage (mean value ± SD) in groups 1 and 2.

Testing Groups	Mean Age, Years	Body Height, cm	Body Weight, kg	BMI, kg/m^2^	Body Fat Mass, %	Lean Body Mass, %
Group 1	13.96 ± 1.4	171.9 ± 9.92	60.6 ± 9.72	20.4 ± 1.45	15.5 ± 4.14	56.7 ± 7.39
Group 2	14.8 ± 1.6	172.6 ± 10.07	63.5 ± 11.94	20.9 ± 2.05	15.6 ± 3.38	58.4 ± 1.95
*p*-Value	0.054	0.827	0.359	0.266	0.973	0.567

**Table 2 nutrients-11-01960-t002:** Comparison of the results obtained by athletes from groups 1 and 2 in 5, 15, and 30 m sprint tests and the standing long jump test (mean value ± SD). No statistically significant difference between the groups was observed.

Testing Groups	5 m Sprint, Seconds	15 m Sprint, Seconds	30 m Sprint, Seconds	Standing Long Jump, Meters
Group 1	1.04 ± 0.07	2.49 ± 0.15	4.45 ± 0.28	2.34 ± 0.17
Group 2	1.06 ± 0.19	2.46 ± 0.16	4.38 ± 0.26	2.38 ± 0.18
*p*-value	0.682	0.382	0.413	0.347

**Table 3 nutrients-11-01960-t003:** Results of 5, 15, and 30 m sprint tests and the standing long jump test in group 1 pre- and post-treatment (mean value ± SD).

Testing Period	5 m Sprint, Seconds	15 m Sprint, Seconds	30 m Sprint, Seconds	Standing Long Jump, Meters
Pre-treatment	1.04 ± 0.07	2.49 ± 0.15	4.45 ± 0.28	2.34 ± 0.17
Post-treatment	1.01 ± 0.06	2.44 ± 0.15	4.35 ± 0.31	2.36 ± 0.19
*p*-value	0.018	0.001	0.016	0.330

**Table 4 nutrients-11-01960-t004:** Body height (cm), body weight (kg), BMI, body fat mass percentage, and lean body mass percentage in group 1 pre- and post-treatment (mean value ± SD).

Testing Period	Body Height, cm	Body Weight, kg	BMI, kg/m^2^	Body Fat Mass, kg	Lean Body Mass, kg
Pre-treatment	171.9 ± 9.92	60.6 ± 9.72	20.4 ± 1.45	15.5 ± 4.14	56.7 ± 7.39
Post-treatment	173.3 ± 8.89	62.6 ± 9.66	20.7 ± 1.62	16.1 ± 4.3	58.18 ± 1.48
*p*-value	<0.001	<0.001	0.008	0.247	0.203

**Table 5 nutrients-11-01960-t005:** Correlations between the change in sprint performance and change in vitamin D and correlations between the change in sprint performance and change in anthropometrics before and after treatment.

Running Speed Tests	Statistics	Vitamin D	Height	BMI
Sprint 5 m	Pearson correlation *p*-value	0.077 0.714	−0.333 0.104	−0.252 0.225
Sprint 15 m	Pearson correlation *p*-value	−0.043 0.84	−0.219 0.292	−0.119 0.571
Sprint 30 m	Pearson correlation *p*-value	0.125 0.553	0.101 0.63	−0.219 0.292
		Weight	Lean body mass	
Sprint 5 m	Spearman correlation *p*-value	−0.369 0.07	0.389 0.05	
Sprint 15 m	Spearman correlation *p*-value	−0.257 0.215	−0.17 0.933	
Sprint 30 m	Spearman correlation *p*-value	−0.05 0.812	0.213 0.297	

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
