# Peer review of "The Dependence of Running Speed and Muscle Strength on the Serum Concentration of Vitamin D in Young Male Professional Football Players Residing in the Russian Federation"

_nutrients, 2019, doi:10.3390/nu11091960_

Round 1
Reviewer 1 Report
This is an interesting manuscript focusing on the role of vitamin D on the athletic performances on young athletes with convincing results. The readership would be benefitted from further clarifications of the following areas:
In the Discussion section, the author should emphasize the limitation of vitamin D supplements, even when it raises serum vitamin D level. The existing studies suggest that serum level is not always reflective to determine the vitamin D status, & consideration should be given on ‘personal vitamin D response index’ (J Steroid Biochem Mol Biol. 2018; 175:12-17); authors should discuss this issue. The author should discuss that vitamin D activity is related to other nutritional factors, including magnesium status of these athletes. Low magnesium could impair vitamin D activities even when the serum level of vitamin D might show adequate, following supplementation (J Am Osteopath Assoc. 2018; 118(3):181-189; Nutrients. 2018; 10(12); DOI: 10.3390/nu10121863). If the investigators have stored serum, might consider measuring magnesium level.Author Response
Comments and Suggestions for Authors
This is an interesting manuscript focusing on the role of vitamin D on the athletic performances on young athletes with convincing results. The readership would be benefitted from further clarifications of the following areas:
In the Discussion section, the author should emphasize the limitation of vitamin D supplements, even when it raises serum vitamin D level. The existing studies suggest that serum level is not always reflective to determine the vitamin D status, & consideration should be given on ‘personal vitamin D response index’ (J Steroid Biochem Mol Biol. 2018; 175:12-17); authors should discuss this issue.
Answer: We agree with the expert reviewer. In the study not all athletes with low vitamin D content had the same response to vitamin D supplementation, which could be due to several factors. One of them is probably the uneven sensitivity of individuals to vitamin D, possibly due to molecular factors.
[Carlberg, C.; Haq, A. The concept of the personal vitamin D response index. Journal of Steroid Biochemical Molecular Biology 2018, 175, 12-17]
The author should discuss that vitamin D activity is related to other nutritional factors, including magnesium status of these athletes. Low magnesium could impair vitamin D activities even when the serum level of vitamin D might show adequate, following supplementation (J Am Osteopath Assoc. 2018; 118(3):181-189; Nutrients. 2018; 10(12); DOI: 10.3390/nu10121863). If the investigators have stored serum, might consider measuring magnesium level.
Answer: We agree with the expert reviewer. However, even a sufficient concentration of vitamin D may not indicate the expected effect on various body functions, since these effects may depend on the content of other biologically active substances in the body, such as magnesium.
[Razzaque, M.S. Magnesium: Are We Consuming Enough? Nutrients 2018, 2, doi: 10.3390/nu10121863]
Reviewer 2 Report
Thank you for the opportunity to review this manuscript submitted to Nutrients. The authors sought to evaluate whether vitamin D status as measured via 25(OH)D is associated with physical performance in young pro football players. I have outlined my primary concerns with the manuscript below, followed by more specific comments and suggestions. I hope the authors find my suggestions and comments to be helpful in improving their work.
GENERAL CONCERNS/COMMENTS
--Throughout the manuscript, the authors are inconsistent with how they refer to vitamin D status. Sometimes they use the generic phrase vitamin D level, while at other times they describe the specific 25(OH)D biomarker. There needs to be consistency in the paper.
--The introduction section it too lengthy and unfocused. I count 10 paragraphs, and this could easily be reduced to 5 or 6 paragraphs.
--The way the authors chose to create groups 1 and 2 is perplexing to me. Why did they leave out athletes within the “reference range” of 30-60 ng/mL? Why compare low (<30) to “excessive” (>60)? Wouldn’t it make more sense to compare athletes that are <30 to those that are 30 and above? Or to compare across 3 groups (<20, 20-30, >30)?
--The lack of control group for the prospective supplementation intervention is a big concern. I understand it may not be ethical to do an RCT with subjects who have a deficiency, but the authors could have at least pre-post tested the athletes within a normal 25(OH)D range and used them as a control group.
ABSTRACT
--pg 1, line 40: Specify the study design in this sentence.
--pg 1, line 40: When the authors say they examined the concentrations of vitamin D, I’m assuming they are referring to 25(OH)D. They need to specific exactly which biomarker they assessed.
--pg 1, line 40: Instead of “…and its effect on running speed…”, I would write “…and its association with running speed…”
--pg 1, lines 42-43: Because there isn’t universal agreement as to what sufficient, deficient, and insufficient levels for vitamin D are (except for bone health), perhaps the authors should put the reference threshold that was used in parentheses after the prevalence numbers.
--pg 1 line 46: When they say 84% were within reference, does that mean above a specific value or within some sort of reference range. Again, this gets back to the issue that the authors didn’t define these thresholds earlier in the abstract.
--pg 2, lines 47-48: Report effect sizes (% change or Cohen d) so that the reader can get a sense of the magnitude of change.
INTRODUCTION
--pg 2, lines 58-60: These sentences contain specific claims and therefore need supportive citations.
--pg 2, line 65: Note sure that “…osteoporosis is reinforced…” is the most optimal wording. Please rephrase.
--pg 2, line 66: The authors should acknowledge that there isn’t universal agreement as to the appropriate cut-off to define adequacy.
--pg 2, line 68: This sentence needs a supportive citation. Furthermore, this sentence is a bit out of place because the remainder of the paragraph mentions nothing about athletes.
--pg 2 lines 78-84: I agree that the issue of vitamin D deficiency is important from a public health standpoint. However, the authors are perhaps over-exaggerating the scope of the problem. When they claim that a billion people are affected worldwide, that is based on thresholds that are fiercely debated. In addition, if vitamin D deficiency is such a pandemic, why does supplementation not reduce the incidence of CVD or cancer (https://www.nejm.org/doi/full/10.1056/NEJMoa1809944)?
--pg 3, line 101: This sentence gives the misleading impression that the literature on vitamin D and muscle is quite strong. Even the citation the authors list [42] states in their concluding paragraph that, “…both the theory and any existence of any ideal 25(OH)D levels needed for peak athletic performance need confirmations by properly conducted interventional trials.”
--pg 3, lines 130-131: Instead of “…and its effect on running speed…”, I would write “…and its association with running speed…”
METHODS
--pg 3, lines 138-142: Was parental consent obtained for athletes that were considered minors?
--pg 3, line 142: Include the average body mass, height, and body fat % in the parentheses as well.
--pg 4, line 147: Define the acronym ARVI.
--pg 4, lines 146-153: I would replace the word participant with the word athlete. The word participant implies someone took part in the study, and these criteria are listing who did not partake in the study.
--pg 4, lines 161-163: This sentence needs supportive citations. I’m particularly interested in the authors claim that >60 ng/ml is considered excessive. What evidence is that claim based on?
--pg 4, lines 164-166: Why are those with values between 30 and 60 ng/ml being omitted from the 2-group analyses?
--pg 4, lines 172-174: How familiarized were the athletes to these tests before the data collection for the study?
--pg 4, lines 178-179: Do the authors mean to say “…with the leading foot located 20 cm before the starting line”?
--pg 4, lines 183-191: The authors probably don’t need to be this detailed when it comes to describing a standing long jump, but I’ll defer to them as to whether they think it’s really necessary.
--pg 5, lines 197-199: Why wasn’t a second set of tests carried out on the vitamin-D-sufficient athletes? Doing so would have provided some sort of control group.
--pg 5, lines 201-202: This section needs more detail. The authors should specify when and how each of these tests was used.
RESULTS
--pg 5, lines 206-208: The text described is not consistent with how the authors said they would report the values in the methods. They are saying here that 75 athletes were within reference range but also say that includes 40 athletes that were above 60 ng/ml. Wouldn’t >60 ng/ml be out of reference range according to how they defined it in the methods section?
--pg 5, lines 209-210: This was already described in the Methods, so it doesn’t need to be repeated here.
--pg 5, lines 210-215: I’m a bit confused about the sample sizes here in comparison to the previous paragraph. In the previous paragraph, the authors said that 56 players were deficient/insufficient and that 40 had excessive levels. However, in this section they are reporting 25 per group. Is that because some athletes didn’t complete the performance tests? If they didn’t complete the performance tests, why are they included in any of the analyses, as not completing the performance tests is listed as an exclusionary criterion on pg 4.
--pg 6, lines 231-234: Why not run correlations between the change in sprint performance and change in vitamin D as well as change in anthropometrics? That could provide some insight as to which one is the more plausible explanation.
--pg 6, Table 1: Please clarify if “Body muscle mass” is the correct phrase, as most body composition assessments quantify lean body mass.
DISCUSSION
--pg 7, line 289: The authors need a weaknesses/limitations section here. Some weaknesses to address include:
Lack of control group for the supplementation D intervention. No assessment of dietary vitamin D intake (which could help explain the prevalence of deficiency in the sample). There is some controversy over the use of 25(OH)D as being the best biomarker to establish deficiency. Further, a lack of standardization in the measurement of 25(OH)D has contributed to confusion in the literature. The authors should discuss some of the limitations of 25(OH)D as a measure of vitamin D status. (See https://bpspubs.onlinelibrary.wiley.com/doi/full/10.1111/bcp.13652)Author Response
Comments and Suggestions for Authors
Thank you for the opportunity to review this manuscript submitted to Nutrients. The authors sought to evaluate whether vitamin D status as measured via 25(OH)D is associated with physical performance in young pro football players. I have outlined my primary concerns with the manuscript below, followed by more specific comments and suggestions. I hope the authors find my suggestions and comments to be helpful in improving their work.
GENERAL CONCERNS/COMMENTS
--Throughout the manuscript, the authors are inconsistent with how they refer to vitamin D status. Sometimes they use the generic phrase vitamin D level, while at other times they describe the specific 25(OH)D biomarker. There needs to be consistency in the paper.
Answer: We agree with the expert reviewer. Given that the serum concentration of precisely 25 (OH) D was determined, it is more correct to use this indicator everywhere
--The introduction section it too lengthy and unfocused. I count 10 paragraphs, and this could easily be reduced to 5 or 6 paragraphs.
Answer: We agree with the expert reviewer. The introduction really contains information widely known to experts in this field and directly not related to the research topic - it was hardly worth repeating. Vitamin D plays a crucial role in phosphorus and calcium metabolism and thus affects the state of bone tissue. It promotes resistance for certain diseases and affects the immune system as well as it maintains muscle tone and the structure of connective tissue. It also regulates lipid and carbohydrate metabolism and the level of blood glucose [1-3]. Vitamin D receptors can be found in a multitude of tissues, which explains its numerous effects outside the skeletal system [4,5]. Chronic vitamin D deficiency in childhood can lead to the development of osteomalacia. Regarding a less severe insufficiency in vitamin D, muscle weakness and balance problems develop, and the process is associated with bone resorption and osteoporosis is reinforced. This may result in an increase in the frequency of fractures. At the same time, adequate vitamin D levels (40 ng/ml and above) are essential for the prevention of fractures, including stress fractures [6-8]. In athletes, maintaining vitamin D levels is important for inflammatory response and speeding up rehabilitation. Vitamin D insufficiency is most prevalent in the regions located to the north of the 35th parallel north, because the sun rays enter the atmosphere at a shallower angle and disperse [9,10]. Most of the vitamin D found in human body is synthesized when UV rays penetrate the open skin at a specific angle. Ensuring an adequate intake of this vitamin only through food is difficult, as its dietary content is quite low [11]. Two distinct forms of vitamin D exist: ergocalciferol (vitamin D2), primarily obtained from plant-based food, and cholecalciferol (vitamin D3), which is synthesized when the body is exposed to UV rays. The vitamin D3 metabolite 25(OH)D is an important agent used in clinical settings to prevent and treat vitamin D deficiencies [12]. Leading population risk factors for vitamin D deficiency are dark skin color, insufficient insolation, obesity, malabsorption syndrome and old age [13-17]. Vitamin D deficiency is currently turning into a pandemic with more than a billion people affected worldwide [18-20]. It is also very common in professional athletes, where it can reach 60% to 90% according to a number of different authors [21-24]. In this population subgroup, the prevalence of Vitamin D deficiency remains high even in regions with sufficient insolation such as Israel, Middle East and Australia, and it reaches 73-90% even during outdoor training. The problem is reinforced by extensive usage of sunscreen, wearing clothes that cover most of the body and training indoors [24-27]. Vitamin D deficiency presents a impressive challenge in football, where it was observed in 64- 83% of football players from England, Spain and Poland [28,29]. However, as shown by Hamilton et al., it is most frequent in the Middle East, where it was diagnosed in 84% of the 342 examined Qatari football players [30]. Vitamin D has been shown to affect muscle tissue, which serves as an important target site. However, most studies confirming the link between vitamin D deficiency and muscle weakness were performed on people of varying ages with no adequate training [11,31-35]. This link may be facilitated through multiple pathways, which either directly affect muscle tissue or, possibly, alter endogenous testosterone synthesis. Animal studies have shown that vitamin D influences myostatin inhibition, regeneration and muscle cell proliferation processes [36,37]. Another possible explanation for the beneficial effect of vitamin D on muscle tissue is that it increases the sensitivity of calcium-binding sites in the sarcoplasmic reticulum, thus strengthening muscle contraction [7]. Pilz et al. showed that vitamin D exhibits an ergogenic potential and indirectly enhances testosterone production, which can also affect the muscular system [38]. It is possible that this is achieved through inhibition of testosterone aromatization and enhanced binding of androgens, which in turn leads to muscle hypertrophy and increase in strength [39-42]. The effects exerted on the muscle tissue by vitamin D fully manifest if the concentration of the vitamin exceeds 50 ng/ml [42]. Notably, most of the existing studies exploring the association between serum vitamin D level and muscle performance in athletes were performed on adults and yielded contradictory results. A meta-analysis by Chiang et al. showed that the administration of vitamin D3 supplements allows achieving a statistically significant improvement in muscle performance, which is not obtained when Vitamin D2 supplements were used [43]. Van Hurst et al. noted that muscle strength and stamina associated with vitamin D administration only occurred in athletes whose vitamin D levels were initially low [44]. Farrokhyar et al. found no association between vitamin D supplementation, vitamin D concentration and various indicators of physical performance, including muscle strength [45]. Football players have also been subjected to such studies. Koundourakis et al. observed a positive correlation between vitamin D level and muscle performance in a cohort of Greek football players [46]. A randomized study performed by Close et al. also showed the beneficial effects of vitamin D on muscle strength and power, as athletes who had received 5,000 IU vitamin D for eight weeks had significantly improved their results in sprint and vertical jump tests [21]. Hamilton et al., on the other hand, found no significant association between the level of 25(OH)D and muscle function [30]. In a study by Jastrzebska et al., where 5,000 IU vitamin D was administered to football players, most of the changes in the indicators of physical performance were insignificant [47]. So far, there have been a few studies regarding the prevalence of vitamin D deficiency in young athletes and of its effect on their muscle performance. Brannstrom et al. observed 19 young female football players and found no statistically significant correlations between vitamin D levels and most of the indicators of muscle tissue performance [48]. A study by Fitzgerald et al. found that vitamin D insufficiency was highly prevalent in a cohort of 53 Canadian junior hockey players. However, no correlation between vitamin D levels and muscle strength was observed [17]. Thus, there is no consensus regarding the effect of serum concentration of vitamin D on running speed and strength in professional athletes. At the same time, most of the existing studies were performed in adult populations, with only occasional studies performed on young athletes. To the best of our knowledge, no published research on this topic performed in a cohort of young football players exists, which increases the importance of a study of the effects exerted on muscle tissue by vitamin D insufficiency. Therefore, the present study examined the serum concentration of vitamin D and its effect on running speed and muscle power in young male football players.
--The way the authors chose to create groups 1 and 2 is perplexing to me. Why did they leave out athletes within the “reference range” of 30-60 ng/mL? Why compare low (<30) to “excessive” (>60)? Wouldn’t it make more sense to compare athletes that are <30 to those that are 30 and above? Or to compare across 3 groups (<20, 20-30, >30)?
Answer: We don’t agree with the expert reviewer. Young football players participated in our study and there was no one over 23 years old. They all ate and trained in the same way, that is, the group was homogeneous and did not begin to share their age, because most of the players were very close in age (about 15 years). Low levels of vitamin D and high levels of vitamin D were compared because we assumed that if vitamin D has any effect, then it will not be significant and the difference in groups with a large difference in serum concentration of 25 (OH) D will be more pronounced in these groups.
--The lack of control group for the prospective supplementation intervention is a big concern. I understand it may not be ethical to do an RCT with subjects who have a deficiency, but the authors could have at least pre-post tested the athletes within a normal 25(OH)D range and used them as a control group.
Answer: We don’t agree with the expert reviewer. The control group was the football players with a high concentration of vitamin D (more than 60), which, according to almost all researchers, is definitely the norm. The players from the high concentration group were tested at the very beginning of the study and their strength and speed parameters were compared with the players from the low vitamin D group.
ABSTRACT
--pg 1, line 40: Specify the study design in this sentence.
Answer: We agree with the expert reviewer. Cohort prospective study
--pg 1, line 40: When the authors say they examined the concentrations of vitamin D, I’m assuming they are referring to 25(OH)D. They need to specific exactly which biomarker they assessed.
Answer: We agree with the expert reviewer. In this situations, clearly indicate 25 (OH) D
--pg 1, line 40: Instead of “…and its effect on running speed…”, I would write “…and its association with running speed…”
Answer: We agree with the expert reviewer. «and its association with running speed…”
--pg 1, lines 42-43: Because there isn’t universal agreement as to what sufficient, deficient, and insufficient levels for vitamin D are (except for bone health), perhaps the authors should put the reference threshold that was used in parentheses after the prevalence numbers.
Answer: We agree with the expert reviewer. Vitamin D levels were below reference in 42.8%(serum 25(ОН)D <30 ng/ml), and above reference in 30.5% of the participants (serum 25(ОН)D 61-130 ng/ml)
--pg 1 line 46: When they say 84% were within reference, does that mean above a specific value or within some sort of reference range. Again, this gets back to the issue that the authors didn’t define these thresholds earlier in the abstract.
Answer: We agree with the expert reviewer. After the treatment, vitamin D concentration increased by 79.2%, and was within reference in 84% of the treated athletes (serum 25(ОН)D 30-60 ng/ml)
--pg 2, lines 47-48: Report effect sizes (% change or Cohen d) so that the reader can get a sense of the magnitude of change.
Answer: We agree with the expert reviewer. Testing was repeated after the end of treatment, and a statistically significant increase in the results of the 5, 15 and 30 m sprint tests was observed (Cohen’s d was 0.46, 0.33, 0.34, respectively), while the results of the standing long jump test remained unchanged.
INTRODUCTION
--pg 2, lines 58-60: These sentences contain specific claims and therefore need supportive citations.
Answer: We agree with the expert reviewer
--pg 2, line 65: Note sure that “…osteoporosis is reinforced…” is the most optimal wording. Please rephrase.
Answer: We agree with the expert reviewer. After reducing the introduction, it will leave the text.
--pg 2, line 66: The authors should acknowledge that there isn’t universal agreement as to the appropriate cut-off to define adequacy.
Answer: We agree with the expert reviewer. After reducing the introduction, it will leave the text.
--pg 2, line 68: This sentence needs a supportive citation. Furthermore, this sentence is a bit out of place because the remainder of the paragraph mentions nothing about athletes.
Answer: We agree with the expert reviewer. After reducing the introduction, it will leave the text.
--pg 2 lines 78-84: I agree that the issue of vitamin D deficiency is important from a public health standpoint. However, the authors are perhaps over-exaggerating the scope of the problem. When they claim that a billion people are affected worldwide, that is based on thresholds that are fiercely debated. In addition, if vitamin D deficiency is such a pandemic, why does supplementation not reduce the incidence of CVD or cancer (https://www.nejm.org/doi/full/10.1056/NEJMoa1809944)?
Answer: We agree with the expert reviewer. After reducing the introduction, it will leave the text.
--pg 3, line 101: This sentence gives the misleading impression that the literature on vitamin D and muscle is quite strong. Even the citation the authors list [42] states in their concluding paragraph that, “…both the theory and any existence of any ideal 25(OH)D levels needed for peak athletic performance need confirmations by properly conducted interventional trials.”
Answer: We agree with the expert reviewer. After reducing the introduction, it will leave the text.
--pg 3, lines 130-131: Instead of “…and its effect on running speed…”, I would write “…and its association with running speed…”
Answer: We agree with the expert reviewer. Therefore, the present study examined the serum concentration of vitamin D and its association with running speed and muscle power in young male football players.
METHODS
--pg 3, lines 138-142: Was parental consent obtained for athletes that were considered minors?
Answer: We agree with the expert reviewer. Consent from the parents of all study participants over 18 years of age was obtained. Without this consent, the local ethics committee would not approve the study.
--pg 3, line 142: Include the average body mass, height, and body fat % in the parentheses as well.
Answer: We agree with the expert reviewer. This study summarizes the data obtained in a cohort of 131 white male football players from Football School Lokomotiv and FC Lokomotiv Moscow Youth team aged 12 to 23 years (mean age 15.6±2.4 years, mean height 172,2±9,9, mean weight 62,1±10,9, mean body fat% 15,6±3,7) who did not have any contraindications for sports
--pg 4, line 147: Define the acronym ARVI.
Answer: We agree with the expert reviewer. The participant suffered from acute respiratory viral infections or any other condition that resulted in absence from three or more training sessions 30 days or fewer prior to the examination;
--pg 4, lines 146-153: I would replace the word participant with the word athlete. The word participant implies someone took part in the study, and these criteria are listing who did not partake in the study.
Answer: We agree with the expert reviewer and changed to:
- The athlete received vitamin D supplements 30 days or fewer prior to first blood sampling;
- The athlete suffered from ARVI or any other condition that resulted in absence from three or more training sessions 30 days or fewer prior to the examination;
- The athlete could not maintain daily contact with the medical personnel distributing vitamin D3 supplements;
- The athlete spent more than three days outside Moscow during the last three months;
- The athlete was expelled from the academy during the study;
- The athlete refused to take part in speed and power testing.
--pg 4, lines 161-163: This sentence needs supportive citations. I’m particularly interested in the authors claim that >60 ng/ml is considered excessive. What evidence is that claim based on?
Answer: We agree with the expert reviewer. This is probably an incorrect translation from Russian into English - we also do not consider the level 25 (OH) D excessive, but consider it simply higher than usual indicators.
--pg 4, lines 164-166: Why are those with values between 30 and 60 ng/ml being omitted from the 2-group analyses?
Answer: We don’t agree with the expert reviewer. We compared only groups with a low level of 25 (OH) D and a high level of 25 (OH) D because we assumed that the concentration of 25 (OH) D could have statistically significant differences only in groups with a large difference in average concentration.
--pg 4, lines 172-174: How familiarized were the athletes to these tests before the data collection for the study?
Answer: We agree with the expert reviewer. All football players have previously repeatedly performed these tests: at least 3-4 times within 2-3 years and are well acquainted with the rules for their conduct.
--pg 4, lines 178-179: Do the authors mean to say “…with the leading foot located 20 cm before the starting line”?
Answer: This is the description of the sprint test we used in this study. The athlete started the sprint from a stationary standing position, with the leading (the nearest leg) foot 20 located cm before the starting line.
--pg 4, lines 183-191: The authors probably don’t need to be this detailed when it comes to describing a standing long jump, but I’ll defer to them as to whether they think it’s really necessary.
Answer: We agree with the expert reviewer. The athlete started the test with the legs shoulder-width apart, feet parallel, arms by their side. Mid-flight, the athlete pulled their legs close to their body, landing on both feet simultaneously. The length of the jump was measured along the perpendicular line from the point of push-off to the athlete's heel upon landing. None of the athletes or personnel performing tests knew which group the athlete was in.
--pg 5, lines 197-199: Why wasn’t a second set of tests carried out on the vitamin-D-sufficient athletes? Doing so would have provided some sort of control group.
Answer: We don’t agree with the expert reviewer. We considered it important to determine changes in strength and speed precisely with a pronounced change in serum concentration of 25 (OH) D. We did not expect significant changes in the concentration of 25 (OH) D in the group with its initially normal concentration, and therefore we did not test this group of athletes.
--pg 5, lines 201-202: This section needs more detail. The authors should specify when and how each of these tests was used.
Answer: We agree with the expert reviewer. IBM SPSS Statistics software v.23.0 (IBM, USA) was used for statistical analysis. Descriptive statistics and Kolmogorov-Smirnov test was used to determine the normality of the distribution. Student’s t-test for independent samples was used to compare weight, height, BMI, body fat mass percentage, 15m sprint in group 1 and 2. Mann-Whitney test for independent samples was used to compare lean body mass, 5m and 15m sprint, standing long jump in group 1 and 2. Student’s t-test for dependent samples was used to compare weight, height, BMI, body fat mass percentage, 30m sprint before and after treatment. Wilcoxon signed-rank test was used to compare 5m, 15m sprint, standing long jump and lean body mass before and after treatment.
RESULTS
--pg 5, lines 206-208: The text described is not consistent with how the authors said they would report the values in the methods. They are saying here that 75 athletes were within reference range but also say that includes 40 athletes that were above 60 ng/ml. Wouldn’t >60 ng/ml be out of reference range according to how they defined it in the methods section?
Answer: We agree with the expert reviewer. In 57.2% (75) young football players, the vitamin D levels were normal (in the range of 30–60 ng / ml in 26.7% (35) and high (61–130 ng / ml in 30.5% (40)
--pg 5, lines 209-210: This was already described in the Methods, so it doesn’t need to be repeated here.
Answer: We agree with the expert reviewer. This sentence must be removed.
--pg 5, lines 210-215: I’m a bit confused about the sample sizes here in comparison to the previous paragraph. In the previous paragraph, the authors said that 56 players were deficient/insufficient and that 40 had excessive levels. However, in this section they are reporting 25 per group. Is that because some athletes didn’t complete the performance tests? If they didn’t complete the performance tests, why are they included in any of the analyses, as not completing the performance tests is listed as an exclusionary criterion on pg 4.
Answer:
At the beginning of the study, the prevalence of vitamin D deficiency was evaluated among young Russian football players - this was the first part of the study. In the second part, changes in strength and speed were analyzed against the background of a significant increase in the concentration of vitamin D- and this part of the study included only football players who passed all the tests - there were 25 people in the group with a deficiency and changes in the concentration of 25 (OH) D and strength and speed evaluated in this group.
--pg 6, lines 231-234: Why not run correlations between the change in sprint performance and change in vitamin D as well as change in anthropometrics? That could provide some insight as to which one is the more plausible explanation.
Answer: We agree with the expert reviewer. Correlations between the change in sprint performance and change in vitamin D and correlations between the change in sprint performance and change in anthropometrics before and after treatment. The ratio of these parameters before and after treatment was used as compared values.
|
|
Vitamin D |
Height |
BMI |
Sprint 5m |
Pearson correlation P-value |
0,077 0,714 |
-0,333 0,104 |
-0,252 0,225 |
Sprint 15m |
Pearson correlation P-value |
-0,043 0,84 |
-0,219 0,292 |
-0,119 0,571 |
Sprint 30m |
Pearson correlation P-value |
0,125 0,553 |
0,101 0,63 |
-0,219 0,292 |
|
|
Weight |
Sprint 5m |
Spearman correlation P-value |
-0,369 0,07 |
Sprint 15m |
Spearman correlation P-value |
-0,257 0,215 |
Sprint 30m |
Spearman correlation P-value |
-0,05 0,812 |
--pg 6, Table 1: Please clarify if “Body muscle mass” is the correct phrase, as most body composition assessments quantify lean body mass.
Answer: We agree with the expert reviewer. This is due to an incorrect translation. It must be written "lean body mass" correctly.
DISCUSSION
--pg 7, line 289: The authors need a weaknesses/limitations section here. Some weaknesses to address include:
Lack of control group for the supplementation D intervention. No assessment of dietary vitamin D intake (which could help explain the prevalence of deficiency in the sample). There is some controversy over the use of 25(OH)D as being the best biomarker to establish deficiency. Further, a lack of standardization in the measurement of 25(OH)D has contributed to confusion in the literature. The authors should discuss some of the limitations of 25(OH)D as a measure of vitamin D status. (See https://bpspubs.onlinelibrary.wiley.com/doi/full/10.1111/bcp.13652)
Answer: We agree with the expert reviewer. There is no “gold standard” that has an exactly estimated the concentration of vitamin D biochemical markers in the human body. Most often, a serum concentration of 25 (OH) D, which is the sum of 25 (OH) D 3 and 25 (OH) D 2, is used for this purpose, although in recent years a number of other biochemical agents have been proposed for this purpose [Sempos, C.RT.; Heijboer, A.C.; Bikle, D.D.; Bollerslev, J.; Bouillon, R.; Brannon, P.M. Vitamin D assays and the definition of hypovitaminosis D: results from the First International Conference on Controversies in Vitamin D. British Journal of Pharmacology 2018, 10, 2194-2207].
At the same time, isotope dilution liquid chromatography mass spectrometry can be considered the most accurate method for determining the biochemical markers of vitamin D, however, current immunnoassys demonstrated acceptable performance [Freeman, J.; Wilson, K.; Spears, R.; Shalhoub, V.; Sibley, P. Performance evaluation of four 25-hydroxyvitamin D assays to measure 25-hydroxyvitamin D2. Clinical Biochemistry 2015, 48,1097-104]. The disadvantages of the study include the lack of a control group for a group with a deficiency of vitamin D, the time of repeated tests for strength and speed. The assessment of vitamin D intake with food was not carried out, however, all athletes ate almost the same way, as they lived and ate together in a boarding school and the effect of vitamin D intake with food was similar among the study participants.
Round 2
Reviewer 2 Report
--pg 2, lines 59-61: Perhaps the authors didn’t understand what I meant by my original comment, but these sentences contain specific claims and therefore need supportive citations. These citations need to be at the end of the sentences that contain the specific claims.
--pg 2, line 69: Replace “…important agent…” with “…important biomarker…”
--pg 2, lines 71-76: Combine these paragraphs into one.
--pg 2, lines 79-81: This sentence about affecting endogenous testosterone needs a supportive citation at the end.
--pg 2, line 85: The Chiang article was a systematic review, not a meta-analysis.
--pg 3, line 107: The use of the word ‘professional’ here is not appropriate, as not all the studies the authors cite used professional athletes.
--pg 3, lines 120-121: I’m not sure the authors addressed my original concern about parental consent. Generally speaking, if children under the age of 18 are to be enrolled, the parent or guardian of the child must provide informed consent on behalf of the child. The authors are now saying that parental consent was obtained for participants over the age of 18. The authors need to be really clear and explicit about how the consent process was handled for those under the age of 18.
--pg 4, lines 163-164: The authors need to take another look at this sentence. The wording is out of order. It should read “…with the leading foot located 20 cm before the starting line” instead of “…with the leading foot 20 located cm before the starting line
--pg 4, lines 185-186: The authors say that 15 m sprint times were compared with both t-tests and Mann Whitney U tests. It can’t be both, so which one is it?
--pg 5, lines 221-222: I appreciate that the authors took my recommendation to examine the correlations between change values in anthropometrics, vitamin D, and performance (Table 5). However, the way they discuss table 5 in the text is unclear. They write “this phenomenon may as well be associated with the increase in anthropometric measurements (Table 5)” but that’s not what the data in Table 5 is showing. Table 5 shows no significant correlations between the changes scores for any of the variables, although there are some trends for weight and height, especially with the 5m sprint. The authors need to reword this section to make it clear what the data in Table 5 show. Also, why did they not examine the correlation between sprint time change and change in lean body mass? Wouldn’t it make physiological sense for increases in lean body mass to correlate with increases in sprint performance?
--pg 7, Table 5: The authors state, “The ratio of these parameters before and after treatment was used as compared values.” I don’t understand what they’re saying with that statement. Please clarify.
--pg 8, lines 294-300: The lack of control group for the prospective part of this study needs to be mentioned as a weakness. I understand that the authors consider the high 25(OH)D group as the control for baseline testing, but because the authors did not track and retest those same subjects during the prospective part of the study, it still means they are missing a longitudinal control group. Why is it important to have a control group for the prospective part of the study? Because the vitamin D supplemented group improved sprint performance, and the authors are unable to determine whether the improvements were due to changes in anthropometrics, 25(OH)D status, or just the fact that these athletes trained for an additional 2 months. If they had a prospective control group, they could potentially tease out some of those things.
Author Response
Comments and Suggestions for Authors
--pg 2, lines 59-61: Perhaps the authors didn’t understand what I meant by my original comment, but these sentences contain specific claims and therefore need supportive citations. These citations need to be at the end of the sentences that contain the specific claims.
Answer: We agree with the expert reviewer. The active form of vitamin D - 1,25(OH)2D increases the efficiency of intestinal calcium absorption from 10%–15% to 30%–40% by interacting with the VDR-RXR (Vitamin D receptor and retinoid X receptor) and thereby promoting the expression of an epithelial calcium channel and a calcium-binding protein [1-3]. It has been estimated that 1,25(OH)2D also increases the intestinal phosphorus absorption from 50%–60% to approximately 80% [4,5].
Holick, M.F. Vitamin D deficiency. Engl. J. Med. 2007, 357, 266–281. Christakos, S.; Dhawan, P.; Porta, A.; Mady, L.J.; Seth, T. Vitamin D and intestinal calcium absorption. Cell. Endocrinol. 2011, 347, 25–29. Christakos, S. Recent advances in our understanding of 1,25-dihydroxyvitamin D3 regulation of intestinal calcium absorption. Biochem. Biophys. 2012, 523, 73–76. Marks, J.; Srai, S.K.; Biber, J.; Murer, H.; Unwin, R.J.; Debnam, E.S. Intestinal phosphate absorption and the effect of vitamin D: A comparison of rats with mice. Physiol. 2006, 91, 531–537. Chen, T.C.; Castillo, L.; Korycka-Dahl, M.; DeLuca, H.F. Role of vitamin D metabolites in phosphate transport of rat intestine. Nutr. 1974, 104, 1056–1060.
--pg 2, line 69: Replace “…important agent…” with “…important biomarker…”
Answer: We agree with the expert reviewer and corrected as suggested
--pg 2, lines 71-76: Combine these paragraphs into one.
Answer: We agree with the expert reviewer and corrected as suggested
--pg 2, lines 79-81: This sentence about affecting endogenous testosterone needs a supportive citation at the end.
Answer: We agree with the expert reviewer. Pilz et al. showed that vitamin D exhibits ergogenic potential and indirectly enhances testosterone production, which can also affect the muscular system. It is possible that this is achieved through inhibition of testosterone aromatization and enhanced binding of androgens, which in turn leads to muscle hypertrophy and increase in strength [27].
Pilz, S., Frisch, S., Koertke, H., Kuhn, J., Dreier, J., Obermayer-Pietsch, B., Wehr, E., Zittermann, A. Effect of vitamin D supplementation on testosterone levels in men. Horm Metab Res. 2011, 43, 223-5.
--pg 2, line 85: The Chiang article was a systematic review, not a meta-analysis.
Answer: We agree with the expert reviewer and corrected as suggested
--pg 3, line 107: The use of the word ‘professional’ here is not appropriate, as not all the studies the authors cite used professional athletes.
Answer: We agree with the expert reviewer.
--pg 3, lines 120-121: I’m not sure the authors addressed my original concern about parental consent. Generally speaking, if children under the age of 18 are to be enrolled, the parent or guardian of the child must provide informed consent on behalf of the child. The authors are now saying that parental consent was obtained for participants over the age of 18. The authors need to be really clear and explicit about how the consent process was handled for those under the age of 18.
Answer: We agree with the expert reviewer. We had to send two types of consent, which were drawn up by the study participants, and not one of them as an example. In Russia, teenagers are very closely monitored for such moments. And they had two types of consent to participate in the study: one for children under 18 and their parents and second – for those over 18. Of course, we will send both forms.
--pg 4, lines 163-164: The authors need to take another look at this sentence. The wording is out of order. It should read “…with the leading foot located 20 cm before the starting line” instead of “…with the leading foot 20 located cm before the starting line
Answer: We agree with the expert reviewer and corrected as suggested
--pg 4, lines 185-186: The authors say that 15 m sprint times were compared with both t-tests and Mann Whitney U tests. It can’t be both, so which one is it?
Answer: We agree with the expert reviewer. We agree with the expert reviewer. We didn’t use both of these tests, it is a misprint. 15 m sprint times were compared with t-tests. Mann Whitney U test was used to compare 5m and 30m sprint times. We corrected this mistake.
--pg 5, lines 221-222: I appreciate that the authors took my recommendation to examine the correlations between change values in anthropometrics, vitamin D, and performance (Table 5). However, the way they discuss table 5 in the text is unclear. They write “this phenomenon may as well be associated with the increase in anthropometric measurements (Table 5)” but that’s not what the data in Table 5 is showing. Table 5 shows no significant correlations between the changes scores for any of the variables, although there are some trends for weight and height, especially with the 5m sprint. The authors need to reword this section to make it clear what the data in Table 5 show. Also, why did they not examine the correlation between sprint time change and change in lean body mass? Wouldn’t it make physiological sense for increases in lean body mass to correlate with increases in sprint performance?
Answer: We agree with the expert reviewer. We examined correlations between change in lean body mass and sprints. Table 5 is shown below:
|
|
Vitamin D |
Height |
BMI |
Sprint 5m |
Pearson correlation P-value |
0,077 0,714 |
-0,333 0,104 |
-0,252 0,225 |
Sprint 15m |
Pearson correlation P-value |
-0,043 0,84 |
-0,219 0,292 |
-0,119 0,571 |
Sprint 30m |
Pearson correlation P-value |
0,125 0,553 |
0,101 0,63 |
-0,219 0,292 |
|
|
Weight |
Lean body mass |
Sprint 5m |
Spearman correlation P-value |
-0,369 0,07 |
0,389 0,05 |
Sprint 15m |
Spearman correlation P-value |
-0,257 0,215 |
-0,17 0,933 |
Sprint 30m |
Spearman correlation P-value |
-0,05 0,812 |
0,213 0,297 |
Thus, increasing blood level of 25 (OH) D in the group of players with an initially insufficient level was associated with a statistically significant increase in performance in 5, 15 and 30 m sprint tests. These changes may be associated not only with an increased level of serum concentration (OH) D, but with a change in anthropometric indicators. However, we did not reveal a correlation between changes in height and weight and sprint rates. We studied the correlation between changes in lean body mass and changes in sprints. It was revealed that only the correlation of lean body mass and the result of sprint by 5m turned out to be significant: the greater the change in lean body mass, the longer the time as a result of running 5m. Also, in the 5m run, there is a tendency for a correlation between changes in weight and height. Other indicators did not significantly correlate with sprint results. (Table 5).
--pg 7, Table 5: The authors state, “The ratio of these parameters before and after treatment was used as compared values.” I don’t understand what they’re saying with that statement. Please clarify.
Answer: We agree with the expert reviewer. There is no need of this proposal. Table 5 shows correlations between changes in sprint times and vitamin D, anthropometrics before and after treatment.
--pg 8, lines 294-300: The lack of control group for the prospective part of this study needs to be mentioned as a weakness. I understand that the authors consider the high 25(OH)D group as the control for baseline testing, but because the authors did not track and retest those same subjects during the prospective part of the study, it still means they are missing a longitudinal control group. Why is it important to have a control group for the prospective part of the study? Because the vitamin D supplemented group improved sprint performance, and the authors are unable to determine whether the improvements were due to changes in anthropometrics, 25(OH)D status, or just the fact that these athletes trained for an additional 2 months. If they had a prospective control group, they could potentially tease out some of those things.
Answer: We agree with the expert reviewer. A group with a high concentration of vitamin D cannot be considered a truly control group, since there were no second measurements in this group. And this is a drawback of the study, which we will indicate in the discussion section.